# The Archer and the Prey: The Duality of PAF1C in Antiviral Immunity

**DOI:** 10.3390/v15051032

**Published:** 2023-04-22

**Authors:** Matthew W. Kenaston, Priya S. Shah

**Affiliations:** 1Department of Microbiology and Molecular Genetics, University of California, Davis, CA 95616, USA; 2Department of Chemical Engineering, University of California, Davis, CA 95616, USA

**Keywords:** polymerase-associated factor 1 complex (PAF1C), antiviral immunity, gene expression, virus–host interactions

## Abstract

In the ongoing arms race between virus and host, fine-tuned gene expression plays a critical role in antiviral signaling. However, viruses have evolved to disrupt this process and promote their own replication by targeting host restriction factors. Polymerase-associated factor 1 complex (PAF1C) is a key player in this relationship, recruiting other host factors to regulate transcription and modulate innate immune gene expression. Consequently, PAF1C is consistently targeted by a diverse range of viruses, either to suppress its antiviral functions or co-opt them for their own benefit. In this review, we delve into the current mechanisms through which PAF1C restricts viruses by activating interferon and inflammatory responses at the transcriptional level. We also highlight how the ubiquity of these mechanisms makes PAF1C especially vulnerable to viral hijacking and antagonism. Indeed, as often as PAF1C is revealed to be a restriction factor, viruses are found to have targeted the complex in reply.

## 1. Introduction

As the first line of host defense, pathogen sensing by the innate immune response culminates in the reconfiguration of host gene expression to restrict virus replication. Polymerase-associated factor 1 complex (PAF1C) plays a critical role in this response. Its diverse molecular functions may bring about death by a thousand cuts for viruses, stemming from PAF1C’s ability to regulate transcription elongation, chromatin remodeling, histone modification, and more.

PAF1C was initially characterized in yeast [1], and the highly conserved complex is composed of PAF1, LEO1, CTR9, RTF1, and CDC73 [2,3]. Concentrated in the nucleus, PAF1C acts as a scaffold to modulate transcription by recruiting histone- and RNA polymerase II (RNAPII)-modifying enzymes to transcription start sites (TSS). The complex coordinates the activity of multiple partners, including elongation factor TFIIS/SI [4], members of the chromatin transcription complex FACT (SPT5, SPT16, POB3) [5], the DRB sensitivity-inducing factor (DSIF) [6], cleavage and polyadenylation specificity factor (CPSF) [7,8], and MYC [9,10]. Moreover, complex members can interact with other transcription factors independently, such as free RTF1 facilitating elongation exclusively [11,12,13]. The diversity of scaffolding partners associated with PAF1C has produced several distinct models for RNAPII pausing, release, and eventual transcript elongation.

PAF1C has been classically viewed as a positive regulator of gene expression, wherein its occupancy at a TSS creates a transcriptional environment conducive to RNAPII release and productive transcription elongation [4,13,14,15,16]. Mechanistically, the SPT6 recruitment of PAF1C likely displaces negative elongation factor (NELF) to mediate RNAPII release and then maintain transcript elongation [17]. However, this model has become more complex in the last decade, with the discovery that RNAPII release is negatively regulated by PAF1C for highly paused genes [10,18,19]. This could be attributed to upstream super-enhancers, where PAF1C occupation potentially blocks RNAPII function, driving the negative regulation of gene expression [18]. Additionally, PAF1C may recruit transcriptional repressors, such as Integrator-PP2A (INTAC) [20] and NELF/SPT5 [21], with a growing number of studies detecting PAF1C-mediated negative regulation [22,23,24,25]. Of course, these distinct mechanisms for transcription are not mutually exclusive, and others have attributed this difference to the modulation of RNAPII velocity and processivity instead of direct suppression [26]. Various molecular switches may also exist to actively reconfigure PAF1C from a transcription attenuator to an elongation factor [27]. Ultimately, the model remains fluid, yet PAF1C remains a key player in modulating gene expression as an RNAPII scaffold. 

Beyond the direct modulation of transcription, PAF1C can affect gene expression indirectly via epigenetic and structural modifications to chromatin, altering accessibility and activity at various TSS. As with transcription, PAF1C relies on several interacting partners to alter histone methylation and ubiquitination [28,29,30,31]. Although this early work was conducted in yeast only, PAF1C interacts with the homologous mammalian complex of proteins associated with Set1 (COMPASS)—composed of SET methyltransferases, myeloid leukemia factors (MLL), and MLL fusion partners—to modulate gene expression [32,33,34,35]. Based on functions in yeast, PAF1C likely modulates DNA replication and damage response, telomere regulation, genome and nucleosome stability, and RNA export [16]. In other words, PAF1C is far from limited to its roles in transcriptional regulation, and its multiplicity of functions makes the complex a keystone of regulating stress responses (Figure 1).

Since the innate immune recognition of viral infections often culminates in an antiviral gene expression program [36], in this review, we will discuss how PAF1C is inherently involved in this process. PAF1C can effectively mount an antiviral response by modulating both interferon and inflammatory gene expression, restricting several distinct virus families. Of course, viruses are notorious for their ability to hinder or hijack host proteins, including the disruption of antiviral signaling [37,38,39]. As methods to identify virus–host interactions are more accessible and widespread [40], we often find that PAF1C is both the archer and the prey when it comes to viruses and innate immunity. 

## 2. Mastermind: PAF1C Controls Antiviral Signaling via Gene Expression

Several omics-based screens have implicated PAF1C in both interferon-related and inflammatory innate immune responses. Both responses are central to antiviral signaling and the ultimate restriction of virus replication. Although work on PAF1C and the immune response is diverse, sometimes incongruous, and complex, these pleiotropic effects may be advantageous to the host under different contexts (Figure 2).

### 2.1. Interferon Response

Upon pathogen detection, type I interferons (IFN-I) initiate broad interferon-stimulated gene (ISG) expression. This contributes to the restriction of viruses in infected cells, inflammation, antigen presentation, and the eventual activation of adaptive immunity [41]. Several studies have implicated PAF1C in the transcriptional modulation of these interferon-related pathways, and they are often seen in the context of specific viruses. For now, we will focus on the elements of these studies that elucidated the unique roles of PAF1C in this response.

Two independent studies have shown drastic decreases in interferon-related gene expression in the absence of PAF1. In one of these studies, the treatment of cells with both poly(I:C) and IFN-β following PAF1 knockdown resulted in a significant reduction in mRNA expression by the RNA-sequencing of known ISGs, including IFITs and OASs [42]. Our own work recapitulated this reduction in immune genes following poly(I:C) treatment in PAF1 knockout cells [43]. More importantly, we paired our PAF1 knockout model with a knockout of STAT2, a transcription factor that transduces the IFN-I signal. While both led to overall decreases in immune gene expression, the affected genes were distinct between knockouts. Gene set enrichment analysis (GSEA) suggested that PAF1 induces DDX58-IFIH1 signaling, including events upstream of IFN-I production, while STAT2 does not. Therefore, PAF1C positively regulates gene expression upstream and downstream of IFN signaling (Figure 2A), a trend shared with the PAF1C modulation of the inflammatory response [44].

However, the disruption of PAF1C by removing individual subunits does not necessarily translate to the full abrogation of an immune response. For example, the overexpression of the CDC73 subunit is sufficient to trigger a IFN-II response via STAT1 activation and full PAF1C complex formation is not required for this to occur [45]. Taken together, there is a clear role for PAF1C in supporting antiviral gene expression upstream and downstream of IFN production, though some mechanisms may rely on individual subunits rather than the entire complex (Figure 2A).

### 2.2. Inflammatory Response

PAF1C subunits were identified as positive regulators of TNF expression—a bellwether of the inflammatory response—through a genome-wide CRISPR screen of primary immune cells following LPS stimulation [46]. PAF1C depletion reduced expression in short- and long-term inflammatory signatures. Additionally, the study recapitulated PAF1C interactions within its own subunits and with additional factors. One novel interacting partner, AU-rich RNA-binding and leucine metabolism protein (AUH), displayed a reduction in TNF levels when knocked down. AUH stabilizes transcripts, and the interaction with several PAF1C subunits suggests that the PAF1C regulation of the inflammatory response is multifaceted (Figure 2B). The complex likely engages in the recruitment of various transcription factors to initiate inflammatory gene expression, but these interactions may also facilitate the further stabilization of the elongation complex. 

In another study exploring the role of PAF1C in the inflammatory response, PAF1C subunits displayed increased expression in mice upon LPS treatment [47]. Using the siRNA knockdown of the CTR9 subunit in mammalian cells and subsequent treatment with both LPS and IL-6, the basal and induced expression of various inflammatory genes were reduced. Additionally, the depletion of CTR9 specifically reduced inflammatory responses for the IL-6 pathway, while having no effect on other inflammatory pathways. The depletion of CTR9 diminished the trimethylation of H3K4 for IL-6 inducible genes and destabilized STAT3 interactions with promoter regions. The overexpression of JAK1 and JAK2 did not compensate for the effects of CTR9 depletion on STAT3 activity, indicating CTR9 is required for proper STAT3 recruitment to target immune genes (Figure 2B). These results suggest PAF1C plays a crucial role in facilitating the inflammatory response by histone methylation and transcription factor recruitment at sites of expression. 

The same research group also investigated PAF1C’s role in IL-1β-mediated inflammatory responses [44]. Silencing the PAF1 subunit increased the expression of genes induced by IL-1β while decreasing the expression of repressed genes. As such, PAF1C apparently counteracts the IL-1β response. PAF1 blocked the productive elongation of paused inflammatory genes downstream of IL-1β, yet histone methylation was not involved in this inhibition. Instead, acetylation at H3K9 and H4K5 increased following IL-1β stimulation in PAF1-deficient cells due to interactions with General Control Non-repressed 5 (GCN5), an acetyltransferase (Figure 2B). Under normal conditions, PAF1C may prevent acetylation and repress IL-1β responses, while GCN5 can displace PAF1C upon immune activation to promote acetylation and subsequent inflammation. 

These results add further complexity to PAF1C-mediated inflammatory responses and emphasize the importance of investigating its precise mechanisms in different contexts. While PAF1C regulates IL-6 and IL-1β responses through inverted mechanisms (Figure 2B), this may enable the unique control of the inflammatory response by recruiting specific transcription factors and histone-modifying enzymes. Such differential effects are likely important for antiviral signaling and may make the viral antagonism of this host factor more difficult. 

### 2.3. Virus Restriction

While our understanding of PAF1C mechanisms continues to mature, the significance of the complex in eliciting an immune response is self-evident. It is no surprise, then, that PAF1C has been repeatedly identified as a restriction factor for several viruses in separate studies. The first of these studies was a whole genome siRNA screen of human immunodeficiency virus 1 (HIV-1) restriction factors [48]. The knockdown of three PAF1C members (PAF1, CTR9, RTF1) significantly enhanced the number of focus forming units detected from a reporter virus in the first screen, with further knockdowns of LEO1 and CDC73 resulting in similar replication phenotypes. Additionally, the knockdown of PAF1 and CTR9 was sufficient to rescue the replication of HIV-2 and simian immunodeficiency virus (SIV) isolates. The increased levels of HIV-1 ssDNA and proviral integration following the knockdown of PAF1C subunits suggested PAF1C is antiviral at the post-entry virus replication stage until integration. In a later study monitoring restriction factor expression in a HIV-1 patient cohort, PAF1 expression was anticorrelated with viremic progressors [49]. In another cohort of patients undergoing antiretroviral therapy, PAF1, CTR9, and RTF1 expression were positively correlated with lower levels of HIV RNA and transcriptional activity [50] (Figure 2C). The relative expression of PAF1 and RTF1 subunits was anticorrelated with HIV-1 infectivity in primary cells as well [51]. Each of these studies underscores how critical PAF1C may be at preventing virus replication and maintaining latency when relevant.

PAF1C also restricts other viruses. The knockdown of PAF1C subunits increased virus replication for influenza A virus (IAV) [42], vesicular stomatitis virus (VSV), Sendai virus (SeV) [45], and herpes simplex virus 1 (HSV-1) [52]. Similar trends were observed for the flaviviruses dengue virus (DENV) [53] and Zika virus (ZIKV) [54], yet the opposite was found for Japanese encephalitis virus (JEV) [54]. Such incongruence may be due to various factors, such as the transient nature of the knockdown systems, differing MOIs, or the context-dependency of PAF1C function in the first place. Yet, it is worth noting DENV exhibited significant increases in replication in PAF1 knockout cells [43]. Overall, these findings connect PAF1C immune regulation to measurable impacts on virus replication across different virus families (Figure 2C), although the context of regulation remains a relevant caveat. 

The means by which this restriction occurs could be through several mechanisms and targets of gene expression. The activation of interferon and inflammatory responses—as previously discussed—is a definite contributor to canonical antiviral signaling, yet there are likely nontraditional pathways of viral restriction given the positive and negative regulatory modes of PAF1C [14,18]. We found PAF1 knockout increased the expression of host dependency factors, indicating that PAF1 typically suppresses these factors [43]. In the context of its opposing functions of pause and release, PAF1C could serve the canonical positive role of inducing ISG expression while suppressing factors positively associated with viral replication. Our recent follow-up study has further addressed this likelihood by treating PAF1 knockout cells with a broad slate of immune stimuli followed by transcriptomics [23]. This included the activation of interferon and inflammatory pathways at both upstream (PAMPs) and downstream levels (IFN-β and TNF-α), allowing us to assess the conserved trends versus the context-dependent nuances of a PAF1C-mediated immune response. 

We broadly captured previously identified PAF1-mediated responses to reinforce several key conclusions. First, PAF1 knockout resulted in up- and down-regulated gene expression, suggesting that PAF1C likely engages in immune activation and suppression across diverse innate immune responses. Secondly, the transcriptional motifs and regulatory elements of these gene groups were distinct. For instance, the genes likely induced by PAF1C were strongly associated with promoter motifs connected to Nuclear factor-κB (NF-κB) signaling and related immune pathways. In contrast, genes expected to be suppressed were weakly associated with promoters and were more enriched for enhancer elements. Since super enhancers are linked to the PAF1C pausing of RNAPII [18,19], this trend suggests a possible mechanism for context-dependent immune gene regulation. Furthermore, repressed genes were enriched for host dependency factors of several viruses, consistent with previous results [43]. While our recent study only evaluated the global transcriptomic landscape, it successfully recapitulated multiple conclusions of past studies on both interferon and inflammatory responses under a single experimental model (Figure 2D). The continued study of PAF1C will prove essential for understanding how the mechanisms of pause, release, and elongation overlap with specific axes of the innate immune response. 

In that same vein, recent work connecting PAF1C-mediated transcription and mitophagy—a form of selective autophagy for mitochondria—provides yet another avenue by which PAF1C can restrict viruses [55]. The study demonstrated that PAF1C is a repressor of mitophagy in mammalian cells, at least for PINK1–PRKN-mediated mitophagy. The knockdown of PAF1 and CTR9 increased the mRNA level of Optineurin (OPTN), a receptor for damaged mitochondria, leading to protein degradation via mitophagy. The overexpression of these subunits, on the other hand, suppressed mitophagy and led to the accumulation of mitochondrial proteins. Autophagic processes, such as mitophagy, are suspected of helping or hindering viruses also restricted by PAF1C [56,57,58,59], and we have already observed genes typically paused by PAF1C to be virus host-dependency factors or immune-related [23,43]. Therefore, studying these processes in the context of PAF1C-mediated gene expression could further contribute to our understanding of the complex’s diverse roles in antiviral programming. 

## 3. Karma: PAF1C Is Vulnerable to Viral Antagonism

PAF1C can facilitate an antiviral response across several axes, sometimes leveraging multiple functions at once. As a positive regulator of transcription elongation, PAF1C can recruit various transcription factors and histone-modifying enzymes to modulate RNAPII pausing, release, and elongation, resulting in the increase or decrease of interferon or inflammatory gene expression, depending on the genetic landscape. All these processes drive both canonical and non-canonical pathways that restrict virus replication (Figure 3A). PAF1C’s antiviral function, in other words, is a unique product of the stimulus, the complex’s interactions with other host factors, and the genomic sites it occupies. It is pleiotropic and virtually ubiquitous in the nucleus. However, here we argue that PAF1C performs its functions all too well, making it a prime target in the great war between virus and host. Unsurprisingly, much of the research we have discussed regarding PAF1C’s role in the innate immune response is a byproduct of studying why viruses target PAF1C at all. 

### 3.1. Influenza Virus: Histone Mimic and Reprogramming SUMOylation

Though not the first study to associate PAF1C with viruses, antagonism by IAV has brought the complex to the forefront of innate immunity in the virus–host arms race. In a seminal study, the NS1 protein of IAV H3N2 was found to possess a histone mimic “tail” that interacted with the PAF1 subunit in a virus strain-specific manner [42]. This interaction was facilitated by the similarity of the NS1 tail to histone H3 and was not observed in H5N1 or H3N2 with truncated versions of NS1. Wild-type IAV exhibited decreased PAF1 and RNAPII levels at transcription end sites (TES) compared to a mutant virus lacking the PAF1 binding sequence. Additionally, more TSS-associated short transcripts were detected, antiviral gene expression was suppressed, and viral titers were higher relative to the mutant virus. These observations collectively confirmed that NS1 prevented PAF1C from facilitating transcription elongation to engage antiviral signaling (Figure 3B). The production of a full transcript for both wild-type and mutant NS1 was measured in vitro to validate that NS1 inhibits elongation. Furthermore, infection by IAV mutants unable to antagonize PAF1 resulted in stronger ISG expression, and these IAV mutants had improved replication in PAF1-depleted cells.

Interestingly, recent studies have correlated PAF1C activity with IAV-induced SUMOylation, an epigenetic regulator of transcription. It has been previously demonstrated that IAV infection increases the rate of SUMOylation [60]. A subsequent study implemented a proteomics-based approach to uncover novel SUMO1/2 substrates during IAV infection [61]. Every subunit of PAF1C was an IAV-induced target, and the shRNA knockdowns of several subunits led to an expected increase in IAV replication. Importantly, the CDC73 subunit relied on SUMOylation to properly stimulate ISG expression, so how does this coincide with PAF1C antagonism? Notably, these trends were proven with a virus incapable of directly targeting PAF1 via a histone mimic. Thus, the mechanism by which IAV antagonizes PAF1C has the potential to be strain specific. Future studies should investigate the two potential modes of antagonism. First, whether the degree of SUMOylation across IAV strains impacts PAF1C-mediated antiviral signaling. Second, whether this is correlated with the likelihood of antagonizing PAF1C through the evolution of a histone mimic on NS1.

### 3.2. Flaviviruses: NS5 Antagonism via Chromatin Interaction

The NS5 protein of flaviviruses is also well-characterized in its ability to interfere with PAF1C-mediated immune responses. Global virus–host proteomics for DENV and ZIKV first revealed that NS5 interacted with several PAF1C subunits [52], with the interaction being subsequently validated for all four DENV serotypes, ZIKV, and West Nile Virus (WNV). Eventually, this interaction was found to be conserved across mosquito- and tick-borne flaviviruses [43,53], suggesting that it plays a consequential role in flavivirus replication. As established previously, the knockdown of various subunits led to increased virus replication (except for JEV). Yet, more importantly, NS5 reduced LEO1 occupancy at DENV-induced ISGs, abrogating the expression of those same genes [52]. It was readily apparent that flavivirus NS5 disrupted PAF1C recruitment to ISGs, but the exact mechanism behind this interference remained unresolved.

Recent work by us and others has addressed this need to characterize the NS5–PAF1C interaction. Our own work established that DENV NS5 relied on nuclear localization to facilitate an interaction with PAF1C [43]. Additionally, the C-terminus region of the NS5 methyltransferase (MTase) domain, referred to as the pivot region, was required for interaction with PAF1C. Upon mutating these regions and eliminating interaction with PAF1C, immune gene expression was significantly rescued, establishing the causal link between this virus–host interaction and the antagonism of antiviral signaling. Another group focusing on ZIKV NS5 expanded our understanding of localization dynamics by discovering NS5’s ability to colocalize and associate with chromatin [62]. Intriguingly, the MTase domain facilitated this chromatin binding, allowing NS5 to repress transcription via the direct occupation of certain TSS, including that of neurodevelopment and immune genes. This mechanism directly resulted in the displacement of PAF1C from several TSS because the strength of NS5-PAF1C and NS5-chromatin interactions exceeds that of PAF1C and chromatin (Figure 3C). 

Connecting these two studies would require discerning whether disruption of the pivot region we isolated may be sufficient to break the competitive binding between NS5, PAF1C, and chromatin. It is also relevant to elucidate whether NS5 antagonism is sufficient to displace PAF1C from paused sites of expression, as current work has focused predominantly on active transcript elongation. Regardless, both studies effectively highlight how flavivirus NS5 quite literally blocks antiviral signaling at PAF1C-mediated sites of expression, boosting virus replication and exacerbating pathogenesis. 

### 3.3. Retroviruses: Exile from the Provirus

Aside from the antiviral signaling outcomes of PAF1C transcriptional regulation, the actual, direct functionality of the complex is especially relevant for the expression of integrated viruses. Following the integration of HIV-1, the transactivator protein, Tat, binds the viral promoter and phosphorylates RNAPII and associated factors to drive the productive elongation of viral transcripts [63]. P-TEFb—part of the super elongation complex (SEC)—is well implicated in this mechanism, so it is unsurprising that every PAF1C member was identified upon the pulldown of HIV-1 Tat protein [64,65]. In a knockdown model, PAF1C and MLL-fusion partners—known as Tatcom1—were required for the transactivation of HIV-1 transcription and basal long terminal repeat (LTR) activity [64] (Figure 4A). Based on these results, Tat actively recruits PAF1 to the viral promoter, and the subsequent formation of Tatcom1 allows for the productive elongation of viral transcripts. Herein, HIV appears to leverage PAF1C function directly, rather than through the antagonism of its antiviral roles. 

However, this model for PAF1C-mediated HIV transcription does not readily align with the maintenance of latency and viral restriction correlated with PAF1C subunits [49,50,51]. How does HIV effectively antagonize and restructure PAF1C-mediated restriction to exit latency? Recent work has uncovered a potential mechanism for reconciling this disparity. Lens epithelium-derived growth factor (LEDGF)/p75—a chromatin-associated factor—facilitates HIV integration by interacting with the viral integrase while repressing proviral transcription during latency [66,67]. In the recent study, LEDGF/p75 interacted with several PAF1C subunits in latent HIV infected cells [32], and the depletion of PAF1—surprisingly—increased the reactivation of HIV transcription. Upon intentionally stimulating the reversal of latency, PAF1 depletion did not impact proviral transcription. Such a connection suggests that LEDGF/p75 associates with PAF1C to maintain latency (Figure 4B), yet PAF1C does not revert to its previous roles in Tatcom1 upon reactivation. Perhaps the early PAF1C-mediated inhibition of HIV is then distinct from its role during latency or is dependent on the genomic context of integrated proviral DNA. To that end, the same study found that RNAPII occupancy increased at HIV genomic sites after PAF1 and LEDGF/p75 depletions, and PAF1 was unable to associate with the LTR without LEDGF/p75 [32] (Figure 4C). Thus, the PAF1C pausing of RNAPII—orchestrated by LEDGF/p75—seems to be the culprit for maintaining HIV latency, highlighting the importance of PAF1C-mediated pausing for viral antagonism. 

The same study articulated how PAF1C is possibly displaced following HIV reactivation. A cascade underlies proviral transcriptional activation during HIV latency reversal: LEDGF/p75 associates and recruits the MLL1 branch of COMPASS to catalyze H3K4 trimethylation [32]. This epigenetic mark promotes SEC binding at the proviral LTR, stimulating proviral initiation and elongation via RNAPII. MLL1 and PAF1 share the same binding site on LEDGF/p75, wherein the forced expression of MLL1 dissociates PAF1 from LEDGF/p75 and vice versa. Although this model recapitulates PAF1C as a restriction factor for HIV, its shortcomings lie in its inability to explain how the HIV provirus hijacks PAF1C during latency. For one, it does not necessarily dispute past work on PAF1C being associated with Tat, for MLL-fusion partners were also necessary for transactivation [64]. Moreover, others have resolved the overall structure of the SEC and found it to be based on a highly flexible and unstructured scaffold [68]. Such flexibility inherently allows for changes in binding affinities, including the dynamic reconfiguration of the binding partners and transcriptional landscape. Therefore, both models of PAF1C and HIV proviral transcription may be distinct manifestations of the same complex.

Tat may contribute to PAF1C displacement in this way, yet other HIV proteins are implicated as well. Recent work found that Vpr expression reduced histone H1.2 ubiquitination, and when H1.2 was affinity purified, proteomics showed Vpr expression enhanced PAF1C binding to H1.2 [69] (Figure 4C). Histone H1 can be integral to chromatin accessibility and the resulting differential gene expression, so Vpr may indirectly antagonize PAF1C via this process to sequester its antiviral functions. Either way, future studies should focus on whether Vpr and Tat are targeting the immune or proviral transcription roles in which PAF1C is involved. It also begs the question as to whether PAF1C displacement and viral activation is a byproduct of host functions over time or a process initiated by the virus during latency. Our convoluted model exemplifies how the relationship between PAF1C and viruses is highly complex, and the functional trade-offs may create a delicate balance between viral hijacking and host restriction. 

Being able to reconfigure this balance to benefit the virus or host could have powerful implications, and the contemporary identification of a small molecule inhibitor, referred to as iPAF1C, makes this a current reality. iPAF1C targets a key binding interface between CTR9 and PAF1 subunits, and the drug disrupted PAF1C chromatin localization and induced RNAPII release [70]. Moreover, the drug’s effect mimicked that of PAF1 depletion across the transcriptomic landscape. Thus, iPAF1C became a candidate for the “kick and kill” HIV therapeutic strategy. This approach forces viral reactivation with latency reversal agents (LRAs) to purportedly clear reactivated cells via cytopathic effects or the host immune response [71,72]. iPAF1C enhanced the activity of existing LRAs in CD4+ T cells because the subsequent release of RNAPII advanced the transcriptional elongation of the provirus [70]. Importantly, iPAF1C alone could reverse latency in human peripheral blood mononuclear cells (PBMCs) derived from persons undergoing ART. Advances such as these reinforce PAF1C as a strong candidate for epigenetic-targeted therapies against viruses [73]. Whether disrupting the complex intentionally or displacing viral antagonists to restore native functions, the pharmacological targeting of PAF1C could modulate gene expression to benefit human health. It also means understanding the complex mechanisms behind the viral hijacking of PAF1C is all the more pertinent. 

### 3.4. DNA Viruses: Another Case of Hijacking

Similar complexities extend to DNA viruses, such as HSV-1. During HSV-1 infection, the transcription of immediate early (IE) genes is a critical rate-limiting step. This step relies on the SEC, including pTEFb, as the inhibition of SEC led to the reduced occupancy of viral IE promoters, less viral transcription, and suppressed viral reactivation as a result [74]. As with HIV Tat, the HSV-1 protein HCF-1—a component of the IE regulatory complex—associates with the SEC, pTEFb, and PAF1C [75]. HCF-1 interactions are likely dynamic, and while interactions with SEC and pTEFb are understood to facilitate RNAPII release, it is unknown whether PAF1C is fulfilling roles in elongation as well or is being displaced from potential pausing status. 

Another IE gene, ICP22, further complicates PAF1C antagonism. As an inhibitor of pTEFb activity [76], ICP22 inhibited RNAPII elongation at the TSS and TES of a model cellular gene [77]. Following immunoprecipitation, such as HCF-1, ICP22 interacted with several transcription elongation factors, including PAF1C and the related FACT complex [77,78]. Considering ICP22 and HCF-1 are predominantly associated with host gene expression [79,80,81] and viral transcription [82], respectively, PAF1C antagonism is potentially doubling: HCF-1 hijacks PAF1C at the provirus while ICP22 stalls it at sites of host gene expression. This may be especially relevant since recent proteomics revealed that PAF1C specifically associates with oligomerized interferon inducible factor 16 (IFI16)—an immune DNA sensor—and colocalizes with ICP4 at HSV-1 genome compartments throughout multiple stages of infection [52]. Although the connection between the IFI16 interaction and ICP4 colocalization is yet to be established as causative, it did correlate with reduced HSV-1 late gene expression. Upon PAF1 knockdown, IE gene expression was unaffected while late gene expression was increased. Therefore, it will be valuable to discern whether the ICP22 targeting of PAF1C attenuates this restriction, affects antiviral sites of expression, or is simply a byproduct of colocalization with viral compartments. Recognizing how these dynamics are shaped by the timeline of DNA virus replication could provide further insights as well. Human adenovirus protein E1A also similarly recruits PAF1C for proviral transcription [83], so fully understanding these recurrent mechanisms is key to discerning how viruses are both similar and distinct in their hijacking of PAF1C. 

### 3.5. SARS-CoV-2: Nucleocapsid as a New Candidate for Study

PAF1C continues to appear in virus–host interaction studies. Most recently, SARS-CoV-2 nucleocapsid (N) has been suggested to interact with PAF1C subunits, yet these interactions were weak and inconsistent across several proteomics-based studies [84,85,86,87]. However, this interaction became more apparent in a concurrent study with the expression and purification of a truncated N (N*) protein expressed differentially across SARS-CoV-2 variants [88]. Upon affinity purification and subsequent mass spectrometry on N*, all PAF1C subunits were clearly identified to be interactors. Additionally, PAF1C was previously implicated in the host regulation of the response to SARS-CoV-2 infection [89,90]. Considering the Alpha, Gamma, and Omicron variants specifically express N* during infection compared to the original strain [39], enrichment for the interaction with PAF1C raises the possibility of a variant-specific mechanism for immune antagonism which evolved over the course of the pandemic. Such a rapid selection for the expression of distinct viral proteins to target PAF1C would underscore how relevant the complex is to antiviral immunity. Further study on N* and its effects on gene expression will allow us to confirm immune antagonism and add SARS-CoV-2 variants to the list of viruses that target PAF1C rather than tolerate it. 

## 4. Conclusions

Despite being able to activate several distinct antiviral regulatory programs through transcriptional and epigenetic controls, PAF1C is the anti-hero of its own story, typically being antagonized or hijacked by the same viruses it restricts. Considering the broad scope of the studies we have discussed, and the sheer evolutionary diversity of viruses implicated, repeated association with PAF1C is far from a coincidence. It is more likely to be an invisible string tying together viruses that evolutionarily converged on targeting such a pleiotropic conserved host factor. Then again, our understanding of the context-based effects of PAF1C, especially as it relates to viruses, is lacking, and new virus–PAF1C interactions continue to be identified contemporarily. Since PAF1C fulfills several functions simultaneously, recognizing which viruses hijack which functions will be crucial to navigating the labyrinth of PAF1C and antiviral immunity.

## Figures and Tables

**Figure 1 viruses-15-01032-f001:**
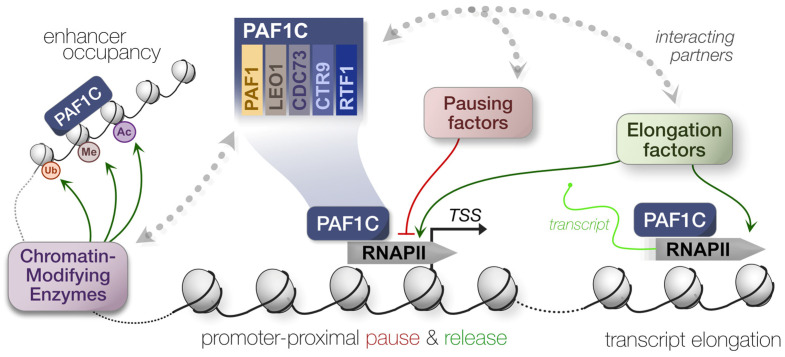
Summary of PAF1C-mediated transcriptional regulation. Polymerase-associated factor 1 complex (PAF1C) engages with several histone- and RNAPII-modifying enzymes to modulate transcription. This can alter the pausing, release, and elongation dynamics of RNAPII.

**Figure 2 viruses-15-01032-f002:**
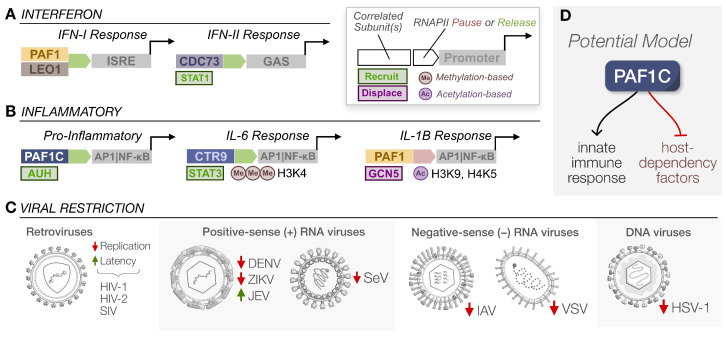
PAF1C-mediated innate immunity and viral restriction. (**A**,**B**) Each graphic summarizes PAF1C subunits implicated in the given type of immune response, whether RNAPII is paused or released, additional factors either recruited or displaced for expression, and commonly associated promoters (not comprehensive) or epigenetic markers. (**C**) PAF1C restricts several viruses across distinct families. (**D**) Integrating existing studies, PAF1C typically mediates innate immune response programs while suppressing factors which may be supportive of virus replication. Note, this model remains incomplete and context-dependent.

**Figure 3 viruses-15-01032-f003:**
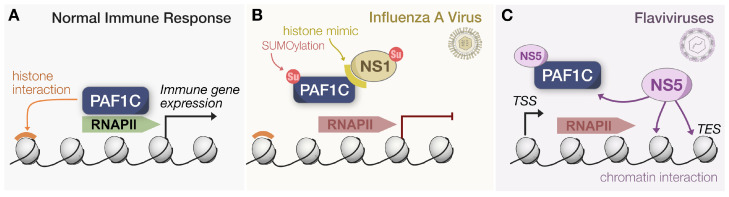
Influenza and Flaviviruses target PAF1C for displacement. (**A**) Under normal conditions, PAF1C relies on chromatin interactions via histones to associate with RNAPII and recruit additional factors for immune gene expression. (**B**) During IAV infection, NS1 translocates to the nucleus. Relying on a histone mimic tail, NS1 can displace PAF1C from RNAPII by competitively binding the complex. Additionally, global SUMOylation increased during IAV infection, and NS1 relies on SUMOylation for functionality during virus replication. (**C**) Flavivirus NS5 localizes to the nucleus and relies on its MTase domain to interact with chromatin. Such an interaction effectively displaces PAF1C from RNAPII during transcript elongation, causing the premature termination of transcription.

**Figure 4 viruses-15-01032-f004:**
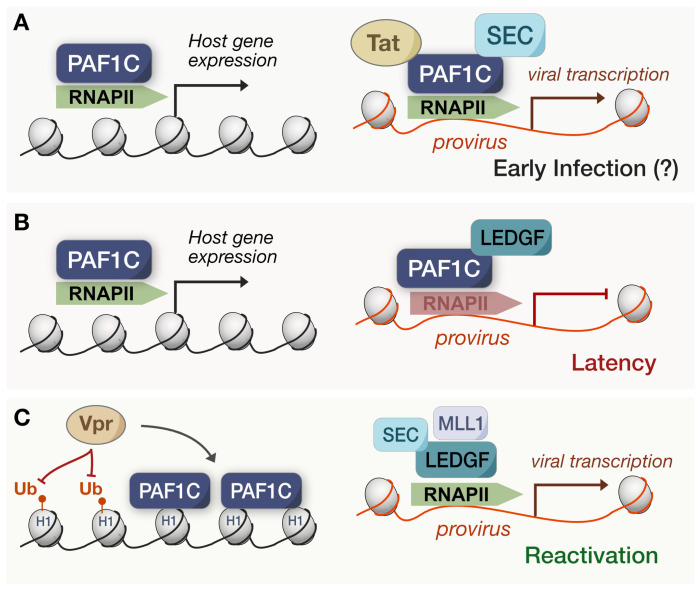
Suggested models of PAF1C in HIV infection. (**A**) In early models, HIV-1 Tat was associated with PAF1C to facilitate proviral expression despite PAF1C maintaining an antiviral state via host gene expression. This early infection model remains questionable. (**B**) During latency, PAF1C partners with LEDGF/p75 to pause RNAPII at the provirus, upholding the latency of the viral infection. (**C**) Upon HIV-1 reactivation, MLL1 displaces PAF1C from LEDGF/p75, allowing for the SEC to mediate the resumed transcription of the provirus. Additionally, HIV-1 Vpr has been associated with reduced H1.2 ubiquitination and promotes the association of PAF1C with these sites.

## Data Availability

Not applicable.

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
