# Peer review of "The Archer and the Prey: The Duality of PAF1C in Antiviral Immunity"

_viruses, 2023, doi:10.3390/v15051032_

Round 1

Reviewer 1 Report

In this review, the roles of PAF1C in regulating gene transcription and hosting innate immune and inflammatory responses were introduced. The authors emphasized the significance and latest findings regarding the involvement of PAF1C in antiviral regulatory processes. Overall, the review provided clear information about the mechanisms of several distinct antiviral regulatory programs, making it a valuable addition to the understanding of PAF1C's role in antiviral immunity.

Reviewer 2 Report

In this review, the authors provide a thorough overview of the implications of PAF1C in the innate immune response against viruses. They begin by giving a background on the functions of the protein in regulating gene transcription, thus providing sufficient context to the reader. The article then moves on to a detailed description of the roles of PAF1C in the transcriptional regulation of the antiviral response, followed by a thorough review of strategies employed by viruses to antagonize or hijack PAF1C.

Overall, the review reads well, is thoughtfully structured, and the figures are well-designed and relevant to the text. The review is comprehensive and meets the objective the authors proposed in their introduction.

Here are some minor suggestions:

1. The authors mentioned a possible role for PAF1C in maintaining/regulating HIV latency. One of the recent papers on this topic was not cited (Soliman, S.H.A. et al., 2023). This paper demonstrated the therapeutic potential of PAF1C for use in HIV latency reversal. Since the authors state in their text and in figures 2 and 4 that PAF1C contributes to latency maintenance, they may want to cite the paper mentioned above.

Full citation: Soliman, S.H.A. et al. (2023) “Enhancing HIV-1 latency reversal through regulating the elongating RNA pol II pause-release by a small-molecule disruptor of PAF1C,” Science Advances, 9(10). 

2. In the section discussing restriction of DNA viruses by PAF1C, a paper with relevant data on effects of PAF1C during HSV-1 infection was not cited (Lum et al., 2019).

Full citation: Lum, K.K. et al. (2019) “Charge-mediated pyrin oligomerization nucleates antiviral IFI16 sensing of herpesvirus DNA,” mBio, 10(4).

3. This review comprehensively discusses antiviral effects of PAF1C through modulating gene expression. Considering the large body of published literature on epigenetic-targeted strategies for treating viral infections (e.g.: “shock and kill” for HIV), it would be insightful to include a paragraph that discusses perspectives on PAF1C’s potential as a target for antiviral therapeutics.

Reviewer 3 Report

This review article discusses PAF1C, a factor that is involved in regulating the body's immune response against viruses. Despite having multiple antiviral functions, PAF1C is often targeted by viruses and used against the body's defense mechanisms. The article suggests that this repeated association is not a coincidence, but rather due to the broad evolutionary diversity of viruses that have converged on targeting this host factor. However, our understanding of the complex effects of PAF1C in relation to viruses is still incomplete, and further research is necessary to identify which functions of PAF1C are targeted by different viruses.

Overall, the article was well-researched, and informative, and provided valuable insights into the topic. I found the arguments presented to be well-supported by the evidence, and the overall structure of the article was easy to follow.

There are a few editing mistakes that need to be corrected.
